Hawksbill × loggerhead sea turtle hybrids at Bahia, Brazil: where do their offspring go?

Proietti Maira C. 1 mairaproietti@gmail.com
Reisser Julia 2 3
Marins Luis F. 4
Marcovaldi Maria A. 5
Soares Luciano S. 6
Monteiro Danielle S. 1 7
Wijeratne Sarath 2
Pattiaratchi Charitha 2
Secchi Eduardo R. 7
1 Instituto de Oceanografia, Universidade Federal do Rio Grande , Rio Grande , Brazil
2 School of Environmental Systems Engineering & Oceans Institute, University of Western Australia , Perth , Australia
3 CSIRO Wealth from Oceans Flagship , Perth , Australia
4 Instituto de Ciências Biológicas, Universidade Federal do Rio Grande , Rio Grande , Brazil
5 Fundação Pró-Tamar , Praia do Forte , Brazil
6 Archie Carr Center for Sea Turtle Research & Department of Biology, University of Florida, Gainesville , FL , USA
7 Núcleo de Educação e Monitoramento Ambiental , Rio Grande , Brazil
Bruno John
Electronic publication date: 2014 Feb 13
Publication date: 2014
Volume: 2
Electronic Location ID: e255
Received 2013 Dec 19; Accepted 2014 Jan 11
Copyright: © 2014 Proietti et al.
Copyright year: 2014
Copyright holder: Proietti et al.
License: This is an open access article distributed under the terms of the Creative Commons Attribution License, which permits unrestricted use, distribution, and reproduction in any medium, provided the original author and source are credited.
License URL: https://creativecommons.org/licenses/by/3.0/

Keywords: Hybridization, Western South Atlantic, Mitochondrial DNA, Juvenile sea turtles, Particle tracking, Dispersal models

Funding: Scholarship and travel funds were provided to MCP by Coordenaçāo de Aperfeiçoamento de Pessoal de Nível Superior (CAPES). Fieldwork and analyses were funded by The Rufford Foundation, Grant #8110-2. JR is sponsored by the International Postgraduate Research Scholarship (IPRS) and CSIROs Flagship Postgraduate Scholarship, and ERS is sponsored by CNPq (307843/2011-4). The funders had no role in study design, data collection and analysis, decision to publish, or preparation of the manuscript.

==============================
Hybridization between hawksbill (Eretmochelys imbricata) and loggerhead (Caretta caretta) breeding groups is unusually common in Bahia state, Brazil. Such hybridization is possible because hawksbill and loggerhead nesting activities overlap temporally and spatially along the coast of this state. Nevertheless, the destinations of their offspring are not yet known. This study is the first to identify immature hawksbill × loggerhead hybrids (n = 4) from this rookery by analyzing the mitochondrial DNA (mtDNA) of 157 immature turtles morphologically identified as hawksbills. We also compare for the first time modeled dispersal patterns of hawksbill, loggerhead, and hybrid offspring considering hatching season and oceanic phase duration of turtles. Particle movements varied according to season, with a higher proportion of particles dispersing southwards throughout loggerhead and hybrid hatching seasons, and northwards during hawksbill season. Hybrids from Bahia were not present in important hawksbill feeding grounds of Brazil, being detected only at areas more common for loggerheads. The genetic and oceanographic findings of this work indicate that these immature hybrids, which are morphologically similar to hawksbills, could be adopting behavioral traits typical of loggerheads, such as feeding in temperate waters of the western South Atlantic. Understanding the distribution, ecology, and migrations of these hybrids is essential for the development of adequate conservation and management plans.

Introduction

Interspecific hybridization occurs naturally or as a result of anthropogenic actions such as habitat modification and fragmentation, species introduction, and population declines (Rhymer & Simberloff, 1996; Allendorf et al., 2001). It is estimated that 25% of plant and 10% of animal species undergo hybridization (Mallet, 2005). This process can contribute to the evolution of many taxa (Barton, 2001), but may also lead to lower fitness and fertility, and even genetic extinction of species (Rhymer & Simberloff, 1996). In the marine environment hybridization has been described for a range of organisms including corals (Willis et al., 2006), fish (Hubbs, 2013), dolphins (Yazdi, 2002), seals (Kovacs, 1997), whales (Glover et al., 2013) and sea turtles (Karl, Bowen & Avise, 1995). Natural hybridization between Cheloniid sea turtle species has been reported for green Chelonia mydas × hawksbill Eretmochelys imbricata, loggerhead Caretta caretta × hawksbill, green × loggerhead, loggerhead × olive ridley Lepidochelys olivacea, and olive ridley × hawksbill turtles (Wood, Wood & Critchley, 1983; Conceição et al., 1990; Karl, Bowen & Avise, 1995; Seminoff et al., 2003; James, Martin & Dutton, 2004; Lara-Ruiz et al., 2006; Reis, Soares & Lôbo-Hajdu, 2010; Vilaça et al., 2012). Possible sterility and lower fitness of these hybrids is concerning since all sea turtle species are currently threatened (IUCN, 2012); however, the exact causes and consequences of these hybridizations are not yet understood.

In Brazil, hawksbill and loggerhead breeding groups present exceptionally high hybridization rates (Lara-Ruiz et al., 2006). The largest rookeries of both species overlap along the coast of Bahia state, where approximately 420 hawksbills and 1240 loggerheads lay their eggs each season (Marcovaldi & Chaloupka, 2007; Marcovaldi et al., 2007). They also overlap temporally, with hawksbills nesting from November to March, and loggerheads from September to February (Marcovaldi & Chaloupka, 2007; Marcovaldi et al., 2007). Studies have shown that 42% of nesting females with hawksbill morphology were actually hybridized with loggerheads, presenting the typical loggerhead mitochondrial DNA (mtDNA) haplotypes BR3 and BR4 (Lara-Ruiz et al., 2006). Since mtDNA is maternally inherited, the first generation (F1) of these hybrids is a cross between female loggerheads and male hawksbills; this could indicate a gender bias since to date no hybrids have presented hawksbill mtDNA (Vilaça & Santos, 2013). This bias has been attributed to the larger loggerhead population and the temporal overlap in nesting at the area. Since the hawksbill season begins around the loggerhead nesting peak (November–December), hawksbill males encounter an abundance of both hawksbill and loggerhead females for mating; meanwhile, by the time a large number of hawksbill females arrive, loggerhead males have already mated and left the area (Vilaça et al., 2012). Interestingly, the hawksbill × loggerhead hybrids are reproductively viable and produce hatchlings, possibly due to an ongoing introgression process (Lara-Ruiz et al., 2006; Vilaça et al., 2012).

After hatching, hawksbill turtles undergo an epipelagic dispersal stage followed by recruitment to tropical coastal areas (Bolten, 2003), usually coral or rocky reefs, where they feed preferably upon incrusting benthic organisms such as sponges and zoanthids (León & Bjorndal, 2002; Proietti, Reisser & Secchi, 2012). Loggerheads also undergo an initial dispersal phase but are adapted to a broader latitudinal distribution range, recruiting to coastal or oceanic areas from tropical to temperate zones, where they feed mainly upon crustaceans, mollusks and fish (Davenport, 1997; Witzell, 2002). Immature loggerhead distribution in Brazil is not well known, but recognized high-use areas include the temperate waters along the southern continental shelf and the Rio Grande rise, a seamount located ca. 800 km off the South of the coast (Bugoni, Krause & Petry, 2003; Monteiro, Bugoni & Estima, 2006; Sales, Giffoni & Barata, 2008). High-occurrence hawksbill feeding areas include the oceanic islands of Rocas Atoll, Fernando de Noronha and São Pedro and São Paulo, and the coastal islands of the Abrolhos National Marine Park (Marcovaldi et al., 1998; Proietti, Reisser & Secchi, 2012). The genetic characterization of hawksbills at these feeding grounds has until now been limited to Rocas Atoll and Fernando de Noronha, and one hybrid individual, representing a hawksbill × loggerhead hybrid backcrossed with a hawksbill (>F1 generation), was found. However it most likely originated from West Africa since it presented an mtDNA haplotype typical of hawksbills from São Tomé and Principe (Monzón-Argüello et al., 2011). Therefore, despite the elevated hybridization between these species in Bahia, how hybrid offspring disperse and where they recruit to is still a mystery. This is likely due to the relatively short timespan of this phenomenon (∼40 years, Lara-Ruiz et al., 2006) and limited surveys at hawksbill and loggerhead feeding grounds.

Understanding how hybridization affects the distribution and ecology of these animals is a complex task that is nevertheless fundamental when defining conservation strategies. In this work, we analyzed mtDNA of 157 immature turtles morphologically identified as hawksbills at high and occasional occurrence areas along the coast of Brazil, and modeled the dispersal patterns of turtles hatched at the Bahia rookery. We report for the first time immature hawksbill × loggerhead hybrids in Brazilian waters and show how temporal variability in hatching period leads to differences between the dispersal patterns of loggerhead, hawksbill, and hybrid offspring from Bahia. Finally, we consider the ecological and conservation implications of this exceptionally frequent phenomenon in Brazil.

Methods

Ethics statement: This work was approved by the evaluation committee of the Biological Oceanography Doctorate Program of the Universidade Federal do Rio Grande. According to Normative Instruction 154/March 2007, all capture, tagging, sampling and transport of biological samples of wild animals for scientific purposes must have approval from Instituto Chico Mendes de Conservação da Biodiversidade (ICMBio) SISBIO committees. This study was approved by the Instituto Chico Mendes de Conservação da Biodiversidade, and conducted under SISBIO licenses #225043, #14122, and #159622. All animal handling was performed by trained personnel, following widely accepted and ethical protocols. When capturing live turtles, the following measures were taken to alleviate stress: (1) turtles were kept out of the water for a maximum of ten minutes; (2) work was performed in a shaded area; and (3) animals were released at the same location of capture.

We analyzed the mtDNA control region of 157 immature turtles morphologically identified as hawksbills from three important Brazilian hawksbill feeding grounds: (1) São Pedro and São Paulo Archipelago (SPSP; n = 12, Curved Carapace Length – CCL = 30–75 cm, mean 53.7 cm); (2) Bahia coast (n = 32, CCL = 21–72 cm, mean 39.7 cm), (3) Abrolhos National Marine Park (n = 65, CCL = 24.5–63.0 cm, mean 37.9 cm); as well as from three areas with sporadic occurrence of this species: (1) Arvoredo Biological Marine Biological Reserve (n = 6, CCL = 30–59.5 cm, mean 41.3 cm); (2) Ceará coast (n = 23, CCL = 22.4–57.5 cm, mean 37.8 cm); and (3) Cassino Beach (n = 25, CCL = 30–60 cm, mean 41 cm; Fig. 1). Loggerheads are not commonly observed at most of these areas (Reisser et al., 2008; Proietti, Reisser & Secchi, 2012), but occur at Ceará (Marcovaldi et al., 2012) and are frequently found at Cassino Beach (Bugoni, Krause & Petry, 2001; Monteiro, Bugoni & Estima, 2006). Samples were collected using disposable scalpels from the flippers of turtles hand-captured in dives at SPSP, Abrolhos, and Arvoredo, and individuals incidentally caught in fishing nets or stranded on beaches (alive or dead) at Ceará, Bahia, and Cassino.

Figure 1 Locations and sample sizes of genetically-described immature hawksbill areas (dots) and the Bahia rookery (red star) in Brazil.

Red dots indicate detection of hawksbill × loggerhead sea turtle hybrids from the Bahia rookery.

Tissue samples were macerated and kept at 37°C in a lysis buffer containing Proteinase K until complete digestion (from 8 to 24 h). DNA was extracted using Genomic DNA Extraction Kits (Norgen Biotek) or the phenol:chloroform method adapted from Hillis et al. (1996). mtDNA control region fragments of approximately 850 bp were amplified via Polymerase Chain Reaction (PCR) using primers LCM15382/H950 (Abreu-Grobois et al., 2006), under the following conditions: denaturation of 5′ at 94°C; 36 cycles of 30′′ at 94°C, 30′′ at 50°C, 1′ at 72°C; final extension of 10′ at 72°C. Illustra GFX purification kits (GE Healthcare) were used for purification, and samples were sequenced in both directions through capillary electrophoresis using an Applied Biosystems® 3130 Genetic Analyzer. Sequences were aligned and cropped to 740 bp using Clustal X 2.0 (Larkin et al., 2007), and classified according to GenBank® and the Atlantic Ocean hawksbill haplotype database (A Abreu-Gobrois, pers. comm., 2013).

Biophysical modeling was performed using the particle-tracking tool ICHTHYOP-3.2 (http://www.previmer.org/en/ichthyop), see model description in Lett et al. (2008) for details. Surface velocity fields were extracted from the global HYbrid Coordinate Ocean Model (HyCOM) with 1/12° reanalysis outputs at daily intervals (http://hycom.org). We chose the fourth-order Runge–Kutta numerical scheme in ICHTHYOP-3.2 to simulate Lagrangian advection of individual particles. The numerical time step was set to 180 s and particle trajectory position outputs were set to daily intervals. Particles were released every 5 days from the Bahia rookery (12–13°S, 37–38°W) proportionally to the monthly amount of hatched loggerheads, hawksbills, and hybrids. Particles were tracked for three years (between May 2009 and June 2013) to encompass the oceanic phase of these sea turtles, following Putman & He (2013).

The monthly proportion of nesting loggerheads and “hawksbills” (including pure and hybrids) were obtained from Marcovaldi & Chaloupka (2007) and Marcovaldi, Vieitas & Godfrey (1999). We then multiplied the monthly number of nesting animals identified as hawksbills (Marcovaldi, Vieitas & Godfrey, 1999) by the monthly percentage of genetically-confirmed hybrid and pure hawksbills (Lara-Ruiz et al., 2006; L. Soares, unpublished data). The hatching periods of loggerheads, hawksbills, and hybrids were calculated by adding 60 days (approximate incubation period; Godfrey et al., 1999; Marcovaldi, Godfrey & Mrosovsky, 1997) to their estimated nesting periods. Finally, the proportion of particles dispersing southwards and northwards was analyzed.

Results

Of the 157 individuals sampled along the coast, four were hawksbill × loggerhead hybrids. Most of these hybrids presented the morphology of pure hawksbill turtles (Fig. 2) and were identified as such, but their mtDNA haplotype was characteristic of nesting loggerheads of the Bahia rookery (BR3). This haplotype was present in one of 23 samples from Ceará (northeast Brazil), and in three of 19 samples from Cassino in the far South (Fig. 1). At Ceará, the hybrid was sampled after being incidentally caught in fisheries, and at Cassino all three hybrids were found dead on the beach. At Cassino one hybrid displayed carapace with overlapping scutes and serrated edges like hawksbills, but a short and thick neck typical of loggerheads (Fig. 2A). This mixed morphology brings additional evidence of this crossbreeding.

Figure 2 Sampled hawksbill × loggerhead sea turtles at Cassino Beach, South Brazil.

Note the relatively large head and thick neck of the individual in A. Photo credits: Nema archive (A, B) and Jonatas H. Prado (C).

Trajectories of simulated virtual particles are shown in Fig. 3. A large proportion of particles moved to the South when released during loggerhead hatching peak (72%; December–March), reaching temperate waters of the western South Atlantic via the Brazil current. Particles released during hybrid hatching peak (January–April) showed a higher southwards displacement (44%) when compared to the hawksbill peak (37%; February –May). Northwards dispersal was higher for particles released during hawksbill (63%), followed by hybrid (56%) and loggerhead (28%) peak hatching seasons.

Figure 3 Virtual particles leaving the Bahia rookery during loggerhead (A), hybrid (B) and hawksbill (C) hatching seasons.

Discussion

In this work we begin to answer a fundamental question that arises when facing the considerable portion of hybrids that nest in Brazil: where do their hatchlings go? Although immature hybrids from the Bahia rookery remain highly undetected relative to the considerable number that is generated, reporting their occurrence at loggerhead feeding grounds (Cassino Beach and Ceará) and their absence at important hawksbill feeding grounds (e.g., Abrolhos, SPSP) is an important step towards better understanding this phenomenon (see Fig. 1). Our modeling approach also highlights the importance of sea turtle nesting season on shaping the spatial distribution of post-hatchlings, with differences observed between hawksbill, loggerhead and hybrid dispersal (see Fig. 3).

While immature hybrids were observed at areas uncommon for hawksbills, they were absent at recognized high-occurrence feeding grounds such as Fernando de Noronha and Abrolhos (this study; Vilaça et al., 2013). Despite the relatively large sample (n = 65) from the tropical reefs of Abrolhos, located very close to the Bahia rookery (ca. 80 km), no hybrids were detected. This could indicate that while these hybrids are morphologically similar to hawksbills, they are not recruiting to the same feeding grounds of pure hawksbills. Three hybrids were found at Cassino Beach, a temperate sandy coast that lacks the optimal characteristics for hawksbill survival (e.g., abundance of preferred food items, relatively high temperatures; Davenport, 1997) and possesses few records of this species (Monteiro, Bugoni & Estima, 2006). Loggerheads on the other hand are commonly found foraging at this region, suggesting that immature hybrids could be adopting the feeding and migration ecology of loggerheads. Similarly, Witzell & Schmid (2003) reported the occurrence of an immature hawksbill × loggerhead hybrid that established its home range in a loggerhead feeding ground.

Adult hawksbill × loggerhead hybrids from Bahia have also been shown to present a distinct ecology when compared to their pure hawksbill counterparts. Marcovaldi et al. (2012) tracked pure hawksbills and hawksbill × loggerhead hybrids after nesting in Bahia and showed different post-nesting migration patterns. Most tracked animals moved along the continental shelf, with all pure hawksbills occupying feeding areas along the eastern coast (Bahia and Alagoas states) while most hybrid females travelled to the northern coast, including Ceará where we detected an immature hybrid. Ceará is an important feeding ground for loggerheads that nest along the coast of Bahia as demonstrated by satellite tracking (Marcovaldi et al., 2010), indicating that the mature female hybrids adopt the behavior of loggerheads. This could also be a possibility for the immature hybrid we detected at the area.

Our biophysical simulations showed that post-hatchling dispersal from Bahia varied according to species: southwards dispersal was proportionally larger throughout loggerhead, followed by hybrid, and lowest during hawksbill peak hatching season. The factors influencing how hybrid sea turtles adopt different feeding and migration behaviors are unknown. Ocean currents influence the dispersal of sea turtle post-hatchlings and are believed to shape the posterior spatial distribution of juveniles and adults (Luschi, Hays & Papi, 2003; Amorocho et al., 2012; Proietti et al., 2012; Putman et al., 2012, 2014; Putman & He, 2013) The model presented here shows that hybrids could have a higher chance of reaching the temperate waters of South Brazil when compared to pure hawksbills. This indicates that these hybrids could already be adopting loggerhead features once they reach the water after hatching. Although pure hawksbills also produce southwards-dispersing hatchlings, they could be limited to lower latitudes by food availability and water temperature, while hybrids could present a behavioral pattern more similar to loggerheads and possibly occupy a wider niche. Another factor that also influences sea turtle dispersal is oriented swimming (e.g., Putman et al., 2012, 2014). For example, if hybrid post-hatchlings navigate mainly southwards while hawksbills swim northwards, the difference in their distribution along the Brazilian coast would be even more pronounced. Further at-sea investigation on hatchling (‘frenzy period’) and post-hatchling swimming behavior (e.g., Thums et al., 2013) is necessary for improving the incorporation of oriented swimming speed and direction in sea turtle post-hatchling dispersal models.

The causes behind the extensive hybridization between hawksbills and loggerheads at the Bahia rookery are still unclear, but could be a result of anthropogenic population declines and uneven population sizes of different species (Lara-Ruiz et al., 2006; Vilaça et al., 2012). It is unknown if this hybridization is threatening the fitness and survival of animals, and the phenomenon should be further investigated for defining whether special measures should be taken when managing these populations. International collaboration might be necessary for determining such management approaches since our particle model shows that ocean currents could transport hybrid turtles from Bahia to distant areas such as Uruguay, Argentina, West African coast, and Western Indian region. Extensive genetic studies in areas of recognized and potential hybrid occurrence, such as loggerhead habitats, are of upmost importance. These studies should combine mtDNA with biparentally-inherited marker analyses for obtaining a better understanding of hawksbill × loggerhead hybrid distribution, parental species and generations. Studies on reproductive and survivorship parameters are also essential for verifying potential negative impacts of this process on long-term viability of local sea turtle populations. Satellite tracking, stable isotopes and diet analyses can also be used to confirm if their movements and feeding habits follow a distinctive pattern. Such studies would provide valuable insight on how the ecology and behavior of sea turtles are affected by hybridization, and consequently guide management practices and strategies to conserve their populations.

Supplemental Information

Supplemental Information 1 Haplotype frequencies of hawksbill turtles at study areas.

Haplotype sequences are shown in the second sheet.

Click here for additional data file.

MCP is a graduate student of the Programa de Pós-graduação em Oceanografia Biológica (FURG). We thank ICMBio, Pata da Cobra Diving, Brazilian Navy, CECIRM PRO-Arquipélago, Abrolhos Park coordination, and all field assistants (a special thanks to B Barbosa) for logistic/field support. We acknowledge Núcleo de Educação e Monitoramento Ambiental (NEMA), Centro de Recuperação de Animais Marinhos (CRAM) and Projeto Tamar for providing samples. This is a contribution of the Research Group ‘Ecologia e Conservação da Megafauna Marinha − EcoMega’.

Additional Information and Declarations

Competing Interests

Author Contributions

Animal Ethics

Field Study Permissions

Julia Reisser is an employee of CSIRO Wealth from Oceans Flagship; Maria A. Marcovaldi is an employee of Fundação Pró-Tamar; Danielle S. Monteiro is an employee of Núcleo de Educação e Monitoramento Ambiental.

Maira C. Proietti, Julia Reisser and Eduardo R. Secchi conceived and designed the experiments, performed the experiments, analyzed the data, contributed reagents/materials/analysis tools, wrote the paper.

Sarath E.M. Wijeratne conceived and designed the experiments, performed the experiments, analyzed the data, contributed reagents/materials/analysis tools.

Maria A. Marcovaldi, Danielle Monteiro and Charitha Pattiaratchi contributed reagents/materials/analysis tools.

Luciano S. Soares analyzed the data, contributed reagents/materials/analysis tools, wrote the paper.

The following information was supplied relating to ethical approvals (i.e., approving body and any reference numbers):

This work was approved by the Biological Oceanography Doctorate Program of the Universidade Federal do Rio Grande.

The following information was supplied relating to ethical approvals (i.e., approving body and any reference numbers):

According to Normative Instruction 154/March 2007, all capture, tagging, sampling and transport of biological samples of wild animals for scientific purposes must have approval from Instituto Chico Mendes de Conservação da Biodiversidade (ICMBio) SISBIO committees.

This study was approved by the Instituto Chico Mendes de Conservação da Biodiversidade, and conducted under SISBIO licenses #225043, #14122, and #159622.

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
