# Peer review of "Hawksbill × loggerhead sea turtle hybrids at Bahia, Brazil: where do their offspring go?"

_PeerJ, doi:10.7717/peerj.255_

## Round 0.1 · original submission · Minor Revisions

Both reviewers liked the science and manuscript. They made a few minor suggestions that could improve it. Could you consider each, make the changes you feel are warranted and explain why (if true) you think a given recommendation is not appropriate.

·

Basic reporting

Overall the manuscript is in a high state of readiness and conforms to reporting requirements. Figures are justified and helpful. The text is tight and conforms to scientific economy.

Typos are few:
206 weather should be whether.
314 Island should be capitalized.
333 K a. should be KA
341 Longline
343 Hawksbill

Experimental design

The experimental design is appropriate. Notably these specimens are very difficult to obtain, and the authors deserve credit for the sampling design. The genetics is conducted correctly and effectively. I am not qualified to evaluate the biophysical modeling, but it does seem pertinent, indicating that seasonal currents may push hybrids into southern feeding areas.

One gap: How were tissue specimens taken from live captures?

Validity of the findings

This paper addresses an interesting phenomenon, the high rate of hybridization among sea turtles in Bahia, Brazil. The report is based on 157 juveniles, putative hawksbills, sampled along the Brazilian coast and offshore islands. While 42% of nesting hawksbills may have hybrid hawksbill/loggerhead origin, the survey only turned up 4 hybrid juveniles, none at the main feeding aggregates for hawksbills. Three of the four were in temperate waters, notably dead strandings, in habitat typical of the more temperate feeding loggerhead.

At my first impression, I thought the theme of hybrids was inappropriate, based on N = 4. However, the Discussion makes a good case that this is a strange finding that could indicate maladaptive behaviors. If the survivorship and behavior of hybrids is the same as pure hawksbills, the survey should have detected about 60 hybrids (out of 157), and so they are significantly below expected abundance. The paper doesn’t answer the question posed in the title, but it documents that this is a compelling question. The paper discovers an interesting phenomenon that is likely to influence research priorities in coming years.

Additional comments

Congratulations to authors on an interesting report.

Reviewer 2 ·

Basic reporting

Pass.

Experimental design

Pass.

Validity of the findings

Pass.

Additional comments

My only hesitation is that it would be nice if additional years could be simulated to capture annual variation in ocean circulation that will influence the proportion of particles moving to the north or south. However, not all published studies have been so robust (e.g., http://link.springer.com/article/10.1007/s00227-011-1712-9), so the authors can choose whether they want to or not.

I would also include more explicit mention (in the discussion) of the possibility that sea turtle swimming behavior could influence predicted results. You already have two papers cited (Putman et al. 2012; 2014) that you could use to support this idea. If for instance, loggerheads typically swim to the south, whereas hawksbills typically swim to the north this could further increase the difference in distribution between hawksbills and hybrids. Regardless, it is very interesting that you see some differences in distribution based on time of hatchling emergence.

The combination of genetics, morphology, and modeling make this a really neat study! I look forward to seeing it published (and citing it!).

---

## Round 0.2 · accepted · Accept

The paper looks great. Thank you for making the suggested edits. Congratulations on the work and thanks for choosing PeerJ.